# PAC-Bayes Compression Bounds So Tight That They Can Explain Generalization

**Sanae Lotfi**[*]     **Marc Finzi**[*]     **Sanyam Kapoor**[*]     **Andres Potapczynski**[*]

**Micah Goldblum**     **Andrew Gordon Wilson**

New York University

## Abstract

While there has been progress in developing non-vacuous generalization bounds for deep neural networks, these bounds tend to be uninformative about why deep learning works. In this paper, we develop a compression approach based on quantizing neural network parameters in a linear subspace, profoundly improving on previous results to provide state-of-the-art generalization bounds on a variety of tasks, including transfer learning. We use these tight bounds to better understand the role of model size, equivariance, and the implicit biases of optimization, for generalization in deep learning. Notably, we find large models can be compressed to a much greater extent than previously known, encapsulating Occam's razor. We also argue for data-independent bounds in explaining generalization.

## 1 Introduction

Despite many more parameters than the number of training datapoints, deep learning models generalize extremely well and can even fit random labels [72]. These observations are not explained through classical statistical learning theory such as VC-dimension or Rademacher complexity which focus on uniform convergence over the hypothesis class [53]. The PAC-Bayes framework, by contrast, provides a convenient way of constructing generalization bounds where the generalization gap depends on the deep learning model found by training rather than the hypothesis set as a whole. Using this framework, many different potential explanations have been proposed drawing on properties of a deep learning model that are induced by the training dataset, such as low spectral norm [57], noise stability [2], flat minima [30], derandomization [55], robustness, and compression [2, 73].

In this work, we show that neural networks, when paired with structured training datasets, are substantially more compressible than previously known. Constructing tighter generalization bounds than have been previously achieved, we show that this compression *alone* is sufficient to explain many generalization properties of neural networks.

In particular:

1. We develop a new approach for training compressed neural networks that adapt the compressed size to the difficulty of the problem. We train in a random linear subspace of the parameters [45] and perform learned quantization. Consequently, we achieve extremely low compressed sizes for neural networks at a given accuracy level, which is essential for our tight bounds. (See Section 4).

2. Using a prior encoding Occam's razor and our compression scheme, we construct the best generalization bounds to date on image datasets, both with data-dependent and data-

---

[*]Equal contribution.

36th Conference on Neural Information Processing Systems (NeurIPS 2022).

independent priors. We also show how transfer learning improves compression and thus our generalization bounds, explaining the practical performance benefits of pre-training. (See Section 5).

3. PAC-Bayes bounds only constrain the adaptation of the prior to the posterior. For bounds constructed with data-dependent priors, we show that the prior alone achieves performance comparable to the generalization bound. Therefore we argue that bounds constructed from data-independent priors are more informative for understanding generalization. (See Section 5.2).

4. Through the lens of compressibility, we are able to help explain why deep learning models generalize on structured datasets like CIFAR-10, but not when structure is broken such as by shuffling the pixels or shuffling the labels. Similarly, we describe the benefits of equivariant models, e.g. why CNNs outperform MLPs. Finally, we investigate double descent and whether the implicit regularization of SGD is necessary for generalization. (See Section 6).

We emphasize that while we achieve state-of-the-art results in both data-dependent bounds and data-independent bounds through our newly developed compression approach, our goal is to leverage these tighter bounds to understand generalization in neural networks. Among others, Figure 1 highlights some of our observations regarding (a) data-dependent bounds, (b) how our method trades-off between data fit and model compression in relation to generalization, and (c) the explanation of several deep learning phenomena through model compressibility using our bounds.

All code to reproduce results is available here.

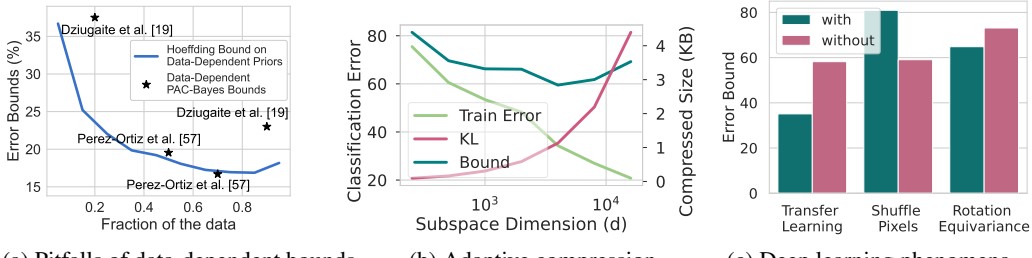

(a) Pitfalls of data-dependent bounds     (b) Adaptive compression     (c) Deep learning phenomena

Figure 1: **The power of data-independent subspace compression bounds for explaining deep learning phenomena.** Bounds for CIFAR-10 except (c)-rotation, which is rotMNIST. **(a)** We show that the simple Hoeffding bound computed *only* on the data-dependent prior and evaluated on the remainder of the training data (essentially measuring validation loss) achieves error bounds that are competitive or even better than data-dependent bounds obtained by previous works, showing that data-dependent PAC-Bayes bounds do not explain generalization any further than the prior alone. Instead, data-independent bounds are more informative for understanding generalization (see Section 5.2). **(b)** Training error, the KL term (compressed model size measured in KB), and our PAC-Bayes bound as the subspace dimension is varied. For a fixed network, our method provides an adaptive compression scheme that trades off compressed size with training error, finding the optimal bound for a given model and dataset. **(c)** We compute our data-independent bounds for model trained *with* and *without*: transfer learning, shuffling the pixels, and the rotation-equivariance property. Our bounds identify the positive impact of transfer learning, how breaking structure in the data by shuffling pixels hurts the model, and that rotationally equivariant models improve generalization on rotated data. Each of these interventions impact the compressibility of the models. See Section 6 for more details.

## 2 Related Work

**Optimizing the PAC-Bayes Bound.** Dziugaite and Roy [18] obtained the first non-vacuous generalization bounds for deep stochastic neural networks on binary MNIST. The authors constructed a relaxation of the Langford and Seeger [41] bound and optimized it to find a posterior distribution that covers a large volume of low-loss solutions around a local minimum obtained using SGD. Ri-

Table 1: **Non-vacuous PAC-Bayes bounds obtained on popular image classification datasets in deep learning.** ⋆ indicates bounds obtained using data-dependent priors (Section 5.2). ✗ indicates that either the method does not support multi-class problems or that it is completely reliant on data-dependent priors and therefore cannot result in data-independent bounds. Additionally, we add Binary MNIST for reference to a benchmark used in earlier works.

| | Non-vacuous PAC-Bayes bounds (%) | | | | | |
|---|---|---|---|---|---|---|
| Reference | Binary MNIST | MNIST | FMNIST | CIFAR-10 | CIFAR-100 | ImageNet |
| Dziugaite and Roy [18] | 16.1 | ✗ | ✗ | ✗ | ✗ | ✗ |
| Rivasplata et al. [60] | 2.2 | ✗ | ✗ | ✗ | ✗ | ✗ |
| Zhou et al. [73] | | 46 | 91.6 | 100 | 100 | 96.5 |
| Dziugaite et al. [19] | | 11⋆ | 38⋆ | 23⋆ | ✗ | ✗ |
| Pérez-Ortiz et al. [59] | | 21.7/1.5⋆ | 49.1 | 90.0/16.7⋆ | 100 | ✗ |
| Our bounds | | **11.6/1.4⋆** | **32.8/10.1⋆** | **58.2/16.6⋆** | **94.6/44.4⋆** | **93.5/40.9⋆** |

vasplata et al. [60] further extended the idea by developing novel relaxations of PAC-Bayes bounds based on Blundell et al. [6].

**Model Compression and PAC-Bayes Bounds.** Noting the robustness of neural networks to small perturbations [29, 30, 40, 39, 37, 57, 8], Arora et al. [2] developed a compression-based approach that uses noise stability. Additionally, they used the ability to reconstruct weight matrices with random projections to study generalization of neural networks. Subsequently, Zhou et al. [73] developed a PAC-Bayes bound that uses the representation of a compressed model in bits, and added noise stability through the use of Gaussian posteriors and Gaussian mixture priors. Furthermore, they achieved even smaller model representations through pruning and quantization [27, 10]. Our compression framing is similar to Zhou et al. [73] but with key improvements. First, we train in a lower dimensional subspace using intrinsic dimensionality [45] and FiLM subspaces [58] which proves to be more effective and adaptable than pruning. Second, we develop a more aggressive quantization scheme with variable length code and quantization aware training. Finally, we exploit the increased compression provided by transfer learning and data-dependent priors.

**Data-Dependent Priors.** Dziugaite et al. [19] demonstrated that for linear PAC-Bayes bounds such as Thiemann et al. [64], a tighter bound can be achieved by choosing the prior distribution to be data-dependent, i.e., the prior is trained to concentrate around low loss regions on held-out data. More precisely, the authors show that the optimal data-dependent prior is the conditional expectation of the posterior given a subset of the training data. They approximate this data-dependent prior by solving a variational problem over Gaussian distributions. They evaluate the bounds for SGD-trained networks on data-dependent priors obtaining tight bounds on MNIST, Fashion MNIST, and CIFAR-10. In a similar vein, Pérez-Ortiz et al. [59] combine data-dependent priors [19] with the PAC-Bayes with Backprop (PBB) [60] to obtain *state-of-the-art* PAC-Bayes non-vacuous bounds for MNIST and CIFAR-10 using data-dependent priors.

**Downstream Transferability.** Ding et al. [16] investigate different correlates of generalization derived from PAC-Bayes bounds to predict the transferability of various upstream models; however, because of this different aim they do not actually compute the full bounds.

Our focus is to achieve better bounds in order to better understand generalization in deep neural networks. For example, we investigate the effects of transfer learning, equivariance, and stochastic training on the bounds, and argue for the importance of data-independent bounds in explaining generalization. We summarize improvements of our bounds relative to prior results in Table 1.

# 3   A Primer on PAC-Bayes Bounds

PAC-Bayes bounds are fundamentally an expression of Occam's razor: simpler descriptions of the data generalize better. As an illustration, consider the classical generalization bound on a finite hypothesis class. Let $\hat{R}(h) = \frac{1}{n}\sum_{i=1}^{n}\ell\left(h\left(x_i\right), y_i\right)$ be the empirical risk of a hypothesis $h \in \mathcal{H}$, with $|\mathcal{H}| < \infty$. Let $\ell$ be the 0-1 loss, and let $R(h) = \mathbb{E}[\hat{R}(h)]$ denote the population risk. With

probability at least $1 - \delta$, the population risk of hypothesis $h$ using $n$ data samples satisfies

$$R(h) \leq \hat{R}(h) + \sqrt{\frac{\log |\mathcal{H}| + \log(1/\delta)}{2n}}. \tag{1}$$

In other words, the population risk is bounded by the empirical risk and a complexity term $\log |\mathcal{H}|$ which counts the number of bits needed to specify any hypothesis $h \in \mathcal{H}$.

But what if we don't believe that each hypothesis is equally likely? If we consider a prior distribution over the hypothesis class that concentrates around likely hypotheses, then we can construct a variable length code that uses fewer bits to specify those hypotheses. Note that if $P$ is a prior distribution over $\mathcal{H}$, then any given hypothesis $h$ will take $\log_2 \frac{1}{P(h)}$ bits to represent when using an optimal compression code for $P$. This prior may result in a smaller complexity term as long as the hypotheses that are consistent with the data are also likely under the prior, regardless of the size of the hypothesis class.

Moreover, the number of bits required can be reduced from $\log_2 \frac{1}{P(h)}$ to $\mathbb{KL}(Q \parallel P)$ by considering a distribution of "good" solutions $Q$. If we don't care which element of $Q$ we arrive at, we can gain some bits *back* from this insensitivity (which could be used to code a separate message). The average number of bits to code a sample from $Q$ using the prior $P$ is the cross entropy $\mathbb{H}(Q, P)$ and we get $\mathbb{H}(Q)$ bits back from being agnostic about which sample $h \sim Q$ to use, yielding the KL-divergence between $Q$ and $P$: $\mathbb{H}(Q, P) - \mathbb{H}(Q) = \mathbb{KL}(Q \parallel P)$.

With these improvements on the finite hypothesis bound — replacing $\log |\mathcal{H}|$ with $\mathbb{KL}(Q \parallel P)$, and sampling a hypothesis $h \in \mathcal{H}$ — we arrive (with minor bookkeeping) at the PAC-Bayes bound introduced in McAllester [51]. This last bound states that with probability at least $1 - \delta$,

$$\mathbb{E}_{h \sim Q}[R(h)] \leq \mathbb{E}_{h \sim Q}[\hat{R}(h)] + \sqrt{\frac{\mathbb{KL}(Q \parallel P) + \log(n/\delta) + 2}{2n - 1}}. \tag{2}$$

Many refinements of Eq. (2) have been developed [41, 50, 7, 64] but retain the same character. That is, the lower the ratio of the KL-divergence to the number of data points $n$, the lower the gap between empirical and expected risk. In this work, we use the tighter Catoni [7] variant of the PAC-Bayes bound (see Appendix J for details).

**Universal Prior.** Leveraging Occam's razor, we can define a prior that explicitly penalizes the minimum compressed length of the hypothesis, also known as the universal prior [62]: $P(h) = 2^{-K(h)}/Z$, where $K$ is the *prefix* Kolmogorov complexity [33] of $h$ (the length of the shortest program that produces $h$ and also delimits itself), and $Z \leq 1$.[2] Using a point mass posterior on a single hypothesis $h^*$, we get the following upper-bound

$$\mathbb{KL}\left(\mathbf{1}_{[h=h^*]} \parallel P(h)\right) = \log \frac{1}{P(h^*)} \leq K(h^*)\log 2 \leq l(h^*)\log 2 + 2\log l(h^*),$$

where $l(h)$ is the length of a given program that reproduces $h$ not including the delimiter. For convenience, we can condition on using the same method for compression and decompression for all elements of the prior. Lastly, we can improve the tightness of the previous PAC-Bayes bound by reducing the compressed length $l(h^*)$ of the hypothesis $h^*$ that we found during training.

**Model Compression.** *Model compression* aims to find nearly equivalent models that can be expressed in fewer bits either for deploying them in mobile devices or for improving their inference time on specialized hardware [9, 10]. For computing PAC-Bayes generalization bounds, however, we only care about the model size. Therefore, we can employ compression methods which may otherwise be unfavorable in practice due to worse computational requirements. *Pruning* and *quantization* are among the most widely used methods for model compression. In this work, we rely on quantization (Section 4.2) to achieving tighter generalization bounds.

## 4 Tighter Generalization Bounds via Adaptive Subspace Compression

Training a neural network involves taking many gradient steps in a high-dimensional space $\mathbb{R}^D$. Although $D$ may be large, the loss landscape has been found to be simpler than typically believed

---

[2]The universal prior similar to the discrete hypothesis prior from Zhou et al. [73] but setting $m(h) = 2^{-2\log_2 l(h)}$ rather than the flat $m(h) = 2^{-72}$.

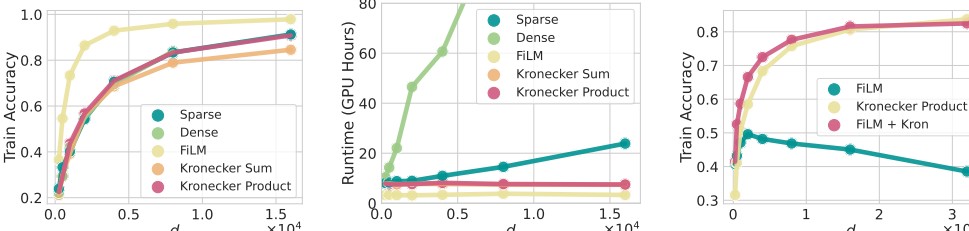

Figure 2: **Effective and scalable projection operators.** **(Left)** Different projection operators $P$ (Section 4.1) used for transfer learning from Imagenet to CIFAR-10 on a ResNet-34 across different subspace dimensions $d$. Kronecker product, Sparse, and Dense perform almost identically **(Center)** Kronecker product runs with substantially reduced the runtime cost compared to the Sparse or Fastfood matrices used by Li et al. [45]. **(Right)** Training from scratch on CIFAR-10. The FiLM projector alone is unable to fit the data when training from scratch, and instead a sum of FiLM and Kronecker Product projectors perform the best.

[14, 26, 23]. Analogous to the notion of intrinsic dimensionality more generally, Li et al. [45] searched for the lowest dimensional subspace in which the network can be trained and still fit the training data. The weights of a neural network $\theta \in \mathbb{R}^D$ are parametrized in terms of an initialization $\theta_0$ and a projection $w \in \mathbb{R}^d$ to a lower dimensional subspace through a fixed matrix $P \in \mathbb{R}^{D \times d}$,

$$\theta = \theta_0 + Pw. \tag{3}$$

To facilitate favorable conditioning during optimization, $P$ is chosen to be approximately orthonormal $P^\top P \approx I_{d \times d}$. For scalability, Li et al. [45] use random normal matrices of the form $P \sim \mathcal{N}(0, 1)^{D \times d} / \sqrt{D}$, and their sparse approximations [46, 43].

In its original form, intrinsic dimensionality (ID) is only used as a scientific tool to measure the complexity of the learning task. Unlike methods like pruning, intrinsic dimension [45] scales with complexity of the task — more complex tasks require a larger intrinsic dimension. Subsequently, we find that ID combined with quantization can serve as an effective model compression method. We note that ideas similar to the intrinsic dimensionality of a model have been explored in the context of model compression for estimating bounds [2]. For our work, the ability to find the intrinsic dimension $d \ll D$ has profound implications for the compressibility of the models, and therefore our ability to construct generalization bounds. As demonstrated by Zhou et al. [73], the compressibility of a neural network has a direct connection to generalization and allows us to compute non-vacuous PAC-Bayes bounds for transfer learning.

Resting upon ID, our key building blocks to achieve tight generalization bounds are composed of (i) a new scalable method to train an intrinsic dimensionality neural network parameterized by Eq. (3) (Section 4.1), and (ii) a new approach to simultaneously train both the quantized neural network weights and the quantization levels for maximum compression (Section 4.2). Our complete method is summarized in Algorithm 1.

## 4.1 Finding Better Random Subspaces

To further improve upon the scalability and effectiveness of the projections $P$ used by Li et al. [45], we introduce three novel projector constructions.

**Kronecker Sum Projector.** Using the Kronecker product $\otimes$, we construct the matrix $P_\oplus = (\mathbf{1} \otimes R_1 + R_2 \otimes \mathbf{1}) / \sqrt{2D}$ where $R_1, R_2 \sim \mathcal{N}(0, 1)^{\sqrt{D} \times d}$ and $\mathbf{1}$ is the vector of all ones in $\mathbb{R}^{\sqrt{D}}$. Noting that $R_1 \perp\!\!\!\perp R_2$ and that the entries are standard normal, $P_\oplus^\top P_\oplus = I_{d \times d} + \mathcal{O}(1/\sqrt{D})$.

**Kronecker Product Projector.** Alternatively, we form the matrix $P_\otimes = Q_1 \otimes Q_2 / \sqrt{D}$ with the smaller $Q_1, Q_2 \sim \mathcal{N}(0, 1)^{\sqrt{D} \times \sqrt{d}}$, and again this matrix is approximately orthogonal: $P_\otimes^\top P_\otimes = (Q_1^\top Q_1 / \sqrt{D}) \otimes (Q_2^\top Q_2 / \sqrt{D}) = I \otimes I + O(D^{-1/4}) = I_{d \times d} + O(D^{-1/4})$. [3]

---

[3]As neither $D$ nor $d$ is typically a perfect square, we concatenate a dense random matrix to pad out the difference between $D$, $d$, and a perfect square. As $(\sqrt{D} + 1)^2 = D + 2\sqrt{D} + 1$, we have that the size of

The matrix vector multiply $w \mapsto Pw$ for both of the above projectors can be performed in time $O(d\sqrt{D})$ and $O(\sqrt{dD})$ respectively, rather than the $O(dD)$ that is required by the dense random matrix. Figure 2 demonstrates the runtime speedup and training performance improvement in comparison to the methods using by Li et al. [45]. Notably, the Kronecker-structured projections retain the fidelity of the dense random matrix while being orders of magnitude faster than the alternative operators when scaling to larger values of $d$. In other words, the Kronecker-structured projectors are as good as the dense projectors for generating random linear subspaces of a given size, but are much scalable.

**FiLM projector.** BatchNorm parameters have an disproportionate effect on the downstream task performance relative to their size. This observation has been used in Featurewise independent Linear Modulation (FiLM) [58, 17] for efficient control of neural networks in many different settings. Several authors have explored performing fine-tuning for transfer learning solely on these parameters and the final linear layer [36]. Drawing on these observations, we construct a projection matrix $P_{\text{FiLM}}$ where only columns corresponding with BatchNorm or head parameters are non-zero and sampled from $\mathcal{N}(0,1)^d/\sqrt{D}$, which we also show in Figure 2. While the FiLM projector is highly effective for transfer learning (shown in Figure 2 left), the performance saturates quickly when training from scratch. For this reason, when training from scratch we employ the sum $P_{\text{FiLM}+\otimes} = (P_{\text{FiLM}} + P_{\otimes})/\sqrt{2}$, which outperforms the two projectors individually as shown in Figure 2 right.

## 4.2 Quantization Scheme and Training

Through quantization, the average number of bits used per parameter can be substantially reduced. When optimizing purely for model size rather than efficiency on specialized hardware, we can choose non-linearly spaced quantization levels which are learned, and use variable length coding schemes as shown in Han et al. [27]. Additionally, the straight through estimator has been central to learning weights in binary neural networks [31]. We combine these ideas to simultaneously optimize the quantized weights and the quantization levels for maximum compression.

Given the full precision weights $w = [w_1, \ldots, w_d] \in \mathbb{R}^d$ and a vector $c = [c_1, \ldots c_L] \in \mathbb{R}^L$ of $L$ quantization levels, we construct the quantized vector $\hat{w} = [\hat{w}_1, \ldots, \hat{w}_d]$ such that $\hat{w}_i = c_{q(i)}$ where $q(i) = \arg\min_k |w_i - c_k|$. The quantization levels $c$ are learned alongside $w$, where the gradients are defined using the straight through estimator [3, 70]:

$$\frac{\partial \hat{w}_i}{\partial w_j} = \delta_{ij} \quad \text{and} \quad \frac{\partial \hat{w}_i}{\partial c_k} = 1_{[q(i)=k]} \tag{4}$$

We initialize $c$ with uniform spacing between the minimum and maximum values in parameter vector $w$ or k-means [11]. To further compress the network, we use a variable length code in the form of arithmetic coding [49], which takes advantage of the fact that certain quantization levels are more likely than others. Given probabilities $p_k$ (empirical fractions) for cluster $c_k$, arithmetic coding of $w$ takes at most $\lceil d \times \mathbb{H}(p) \rceil + 2$ bits, where $\mathbb{H}(p)$ is the entropy $\mathbb{H}(p) = -\sum_k p_k \log_2 p_k$. For a small number of quantization levels, arithmetic coding yields better compression than Huffman coding.

In summary, we use $\lceil d \times \mathbb{H}(p) \rceil + 2$ bits for coding the quantized weights $\hat{w}$, $16L$ bits for the codebook $c$ (represented in half precision), and additional $L \times \lceil \log_2 d \rceil$ bits for representing the probabilities $p_k$, arriving at $l(w) \leq \lceil d \times \mathbb{H}(p) \rceil + L \times (16 + \lceil \log_2 d \rceil) + 2$. As we show in Appendix B, we optimize over the subspace dimension $d$ and the number of quantization levels $L$ and any other hyperparameters, by including them in the compressed description of our model, contributing only a few extra bits.

## 4.3 Transfer Learning

For transfer learning, we replace $\theta_0$ with a learned initialization $\theta_{\mathcal{D}}$ that is found using the pretraining task and data $\mathcal{D}$. With the ID compression, the universal prior $P(h \mid \theta_{\mathcal{D}}) \propto 2^{-K(h|\theta_{\mathcal{D}})}$ will place higher likelihood on solutions $\theta$ that are close to the pre-training solution $\theta_{\mathcal{D}}$.

---

this padding is at most $\sqrt{D} \times \sqrt{d}$, so it does not increase the asymptotic cost of performing the matrix vector multiplies.

---

**Algorithm 1** Compute PAC-Bayes Bound.

---

1: **Inputs:** Neural network $f_\theta$, Training dataset $\{x_i, y_i\}_{i=1}^n$, Clusters $L$, Intrinsic dimension $d$, Confidence $1 - \delta$, and Prior distribution $P$.
2: **function** COMPUTE_BOUND($f_\theta, L, d, (x_i, y_i)_{i=1}^n, \delta, P$)
3:     $w \leftarrow$ TRAIN_ID($f_\theta, d, (x_i, y_i)_{i=1}^n$)                                    ▷ (Section 4.1)
4:     $\hat{w} \leftarrow$ TRAIN_QUANTIZE($w, L, (x_i, y_i)_{i=1}^n$)
5:     Compute quantized train error $\hat{R}(\hat{w})$.
6:     $\mathbb{KL}(Q, P) \leftarrow$ GET_KL($\hat{w}, P$)                                    ▷ (Section 3)
7:     **return** GET_CATONI_BOUND($\hat{R}(\hat{w}), \mathbb{KL}(Q, P), \delta, n$)          ▷ (Section 3)
8: **end function**
9: **function** TRAIN_QUANTIZE($w, L, (x_i, y_i)_{i=1}^n$)                              ▷ (Section 4.2)
10:     Initialize $c \leftarrow$ GET_CLUSTERS($w, L$)
11:     **for** $i = 1$ to quant_epochs **do**
12:         $c \leftarrow c - \rho \nabla_c \mathcal{L}(w, c)$ and $w \leftarrow w - \rho \nabla_w \mathcal{L}(w, c)$
13:     **end for**
14:     **return** $\hat{w}$
15: **end function**
16: **function** GET_KL(($\hat{w}, P$))
17:     $c$, count $\leftarrow$ GET_UNIQUE_VALS_COUNTS($\hat{w}$)
18:     message_size $\leftarrow$ DO_ARITHMETIC_ENCODING($\hat{w}, c$, count)
19:     message_size $\leftarrow$ message_size + hyperparam_search             ▷ (Appendix B)
20:     **return** message_size $+ 2 \times \log($message_size$)$
21: **end function**

---

# 5 Empirical Non-Vacuous Bounds

Combining the training in structured random subspaces with our choice of learned quantization, we produce extremely compressed but high performing models. Using the universal prior, we bound the generalization error of these models and optimize over the degree of compression via the subspace dimension and other hyperparameters as summarized in Algorithm 1. We additionally describe hyperparameters, architecture specifications for each experiment, and other experimental details in Appendix E. In the following subsections, we apply our method to generate strong generalization bounds in the data-independent, data-dependent, and transfer learning settings.

## 5.1 Non-Vacuous PAC-Bayes Bounds

We present our bounds for the *data-independent* prior in Table 2. We derive the first non-vacuous bounds on FashionMNIST, CIFAR-10, and CIFAR-100 without data-dependent priors. These results have particular significance, as we argue in Section 5.2 that using data-dependent priors are not explanatory about the learning process. In particular, we improve over the compression bound results obtained by Zhou et al. [73] on MNIST from $46\%$ to $11.55\%$ and on ImageNet from $96.5\%$ to $94.1\%$. In terms of compression, we dramatically improve the rates as we reduce the compressed size for the best MNIST bound by $94\%$ bringing it down from 6.23 KB to 0.38KB with LeNet5 and, on ImageNet, by $87\%$ bringing it down from 358 KB to 46.3 KB with MobileViT. Since we perform transfer learning with an ImageNet-trained checkpoint, we omit transfer learning experiments on the ImageNet (downstream) dataset. The tightness of our SOTA subspace compression bounds allows us to improve the understanding of several deep learning phenomena as discussed in Section 6. See Appendix E.1 for model architectures and Appendix A for additional results.

## 5.2 Data-Dependent PAC-Bayes Bounds

So far, we demonstrated the strength of our bounds on *data-independent* priors, where we considerably improve on the state-of-the-art. However, a number of recent papers have considered data-dependent priors as a way of achieving tighter bounds [59, 19]. In this setup, the training data $\mathcal{D} = \{(x_i, y_i)\}_{i=1}^n$ is partitioned into two parts, $\mathcal{D}_a$ and $\mathcal{D}_b$, with length $n - m$ and $m$. The first dataset is used to construct a data-dependent prior $P(h \mid \mathcal{D}_a)$, and then the bound is formed over

Table 2: **Our PAC-Bayesian subspace compression bounds compared to *state-of-the-art* (SOTA) bounds.** All results are with $95\%$ confidence, i.e. $\delta = .05$. The sign † refers to data-independent SOTA numbers that we computed using [59], which we run on the additional datasets.

| Dataset | Data-independent priors | | Data-dependent priors | |
|---|---|---|---|---|
| | Err. Bound (%) | SOTA (%) | Err. Bound (%) | SOTA (%) |
| MNIST | **11.6** | 21.7 [59] | **1.4** | 1.5 [59] |
| + SVHN Transfer | **9.0** | 16.1† | | |
| FashionMNIST | **32.8** | 46.5† | **10.1** | 38 [19] |
| + CIFAR-10 Transfer | **28.2** | 30.1† | | |
| CIFAR-10 | **58.2** | 89.9† | **16.6** | 16.7 [59] |
| + ImageNet Transfer | **35.1** | 54.2† | | |
| CIFAR-100 | **94.6** | 100† | **44.4** | – |
| + ImageNet Transfer | **81.3** | 98.1† | | |
| ImageNet | **93.5** | 96.5 [73] | **40.9** | – |

the remaining part of the process: the adaptation of the prior $P(h \mid \mathcal{D}_a)$ to the posterior $Q(h)$ using the data $\mathcal{D}_b$. The empirical risk is computed over $\mathcal{D}_b$ only. Intuitively, using dataset $\mathcal{D}_a$ it is possible to construct a much tighter prior over the possible neural network solutions. In our setting, similar to transfer learning, we use the prior $P_{\mathcal{D}_a}(\theta) = 2^{-K(\theta|\theta_{\mathcal{D}_a})}/Z$ where for compression we use $\theta = \theta_{\mathcal{D}_a} + Pw$, and $\theta_{\mathcal{D}_a}$ is the solution found by training the model (without random projections) on the data $\mathcal{D}_a$ rather than initializing randomly. With these data-dependent priors, we achieve the best bounds in Table 2.

However, our adaptive approach exposes a significant downside of data-dependent priors. To the extent that PAC-Bayes bounds can be used for explanation, data-dependent bounds only provide insights into the procedure used to adapt the prior $P_{\mathcal{D}_a}(\theta)$ to the posterior $Q$ using $\mathcal{D}_b$: any learning that is done in finding $P_{\mathcal{D}_a}(\theta)$ is not constrained or explained by the bound. Given the ability to adapt the size of the KL to the difficulty of the problem, it is possible to squeeze all of the learning into $P_{\mathcal{D}_a}(\theta)$ and none in this adaption to $Q$. This phenomenon happens as the $\mathbb{KL} \to 0$, which we find happens empirically (or very nearly so) across splits of the data, and especially when $n - m$ is large. Setting $Q(\theta) = \mathbf{1}_{[\theta = \theta_{\mathcal{D}_a}]}$, the KL has only the contribution from the optimization over $d$: $\mathbb{KL}(Q||P_{\mathcal{D}_a}) \leq \log D$. We find that the bound is nothing more than a variant of the simple Hoeffding bound where $\mathcal{D}_b$ is the validation set $R\left(\theta_{\mathcal{D}_a}\right) \leq \hat{R}_{\mathcal{D}_b}\left(\theta_{\mathcal{D}_a}\right) + \sqrt{\frac{\log(Dm/\delta)+2}{2m-1}}$.

We can see this phenomenon in Figure 1(a) where we compare existing data-dependent bounds to the simple Hoeffding bound applied directly to the data-dependent prior which was trained on only a small fraction of the data. We can consider the Hoeffding bound as the simplest data-dependent bound without any fine-tuning so that the *prior*, a single pre-trained checkpoint, is directly evaluated on held-out validation data with no KL-divergence term. If another data-dependent bound cannot achieve significantly stronger guarantees than the prior Hoeffding bound, then it only explains that neural networks generalize because the priors already have low validation error which is no explanation for generalization at all. Indeed, we see in Figure 1 that the strength of existing data-dependent bounds relies almost entirely on the a priori properties of the data-dependent prior rather than constraining the actual learning process through compressibility. Similarly, from a minimum description length (MDL) perspective, data-independent bounds can be used to provide a lossless compression of the training data, whereas data-dependent bounds cannot (see Appendix H).

We also note that with data-dependent priors, optimization over the subspace dimension selects very low dimensionality, even if the data does not have low intrinsic dimension. Because most of the data fitting is moved into fitting the prior, the bound selects a low complexity solution with respect to the prior without hurting data fit by choosing a low subspace dimensionality (Appendix D).

By contrast, data-independent bounds explain generalization for the entirety of the learning process. Similarly, our transfer learning bounds meaningfully constrain what happens in the fine-tuning on the downstream task, but they do not constrain the prior determined from the upstream task.

## 5.3 Non-Vacuous PAC-Bayes Bounds for Transfer Learning

By directly interpreting PAC-Bayes bounds through the lens of compression, we immediately see the benefits of using an upstream dataset for transfer learning. Transfer learning allows us to constrain the prior $P(\theta \mid \theta_{\mathcal{D}_a})$ around parameters consistent with the upstream dataset $\mathcal{D}_a$, reducing the KL-divergence between the prior and the posterior and leading to even tighter bounds as we show in Table 2. Our tighter data-independent transfer learning bounds provide a theoretical certification that transfer learning can improve generalization. Our PAC-Bayes transfer learning approach also indicates that transfer learning can boost generalization whenever codings optimized on a pre-training task are more efficient for encoding a downstream posterior than an a priori guess made before seeing data. By contrast, downstream tasks which greatly differ from the upstream task may only be consistent with models that are not compressible under the learned prior, a scenario that describes negative transfer. See Appendix C for more details.

## 6 Understanding Generalization through PAC-Bayes Bounds

The classical viewpoint of uniform convergence focuses on properties of the hypothesis class as a whole, such as its size. In contrast, PAC-Bayes shows that the ability to generalize is not merely a result of the hypothesis class but also a result of the particular dataset and the characteristics of the individual functions in the hypothesis class. After all, many elements of our hypothesis class are not compressible, yet in order to guarantee generalization, we choose the ones that are. Real datasets actually contain a tremendous amount of structure, or else we could not learn from them as famously argued by Hume [32] and No Free Lunch theorems [67, 25]. This high degree of structure in real-world datasets is reflected in the compressibility of the functions (i.e. neural networks) we find in our hypothesis class which fit them.

In this section, we examine exactly how dataset structure manifests in compressible models by applying our generalization bounds, and we see what happens when this structure is broken, for example by shuffling pixels or fitting random labels. Corrupting the dataset degrades both compressibility and generalization.

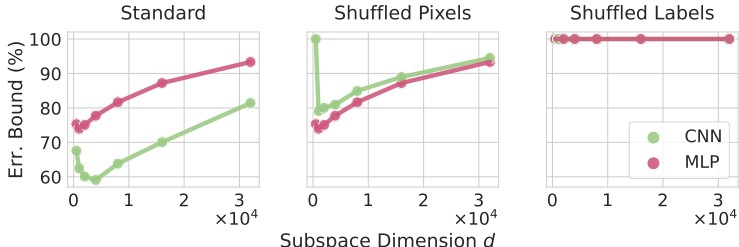

Figure 3: **Breaking structure in the data and the model**. Our PAC-Bayes bound computed using various subspace dimensions for a fixed size CNN and MLP, both with 500k parameters. We train on **(left)** CIFAR-10, **(center)** CIFAR-10 with shuffled pixels, **(right)** CIFAR-10 with shuffled labels. Structure in the dataset induces structure in the model. As structure is removed from the dataset, models which fit the data become much less compressible, hence generalize worse.

**MLPs vs CNNs.** It is well known that convolutional neural networks (CNNs) generalize much better than standard multilayer perceptrons (MLPs) with alternating fully-connected layers and activation functions on image classification problems, even when controlling for the number of parameters. In our generalization bounds, this is reflected in the improved compressibility of CNNs when compared to MLPs. In Figure 3 (left), we see how on CIFAR-10 a we are able to find a lower description length for a CNN than an MLP with the same number of parameters. See Appendix E.4 for experimental details.

**Shuffled Pixels.** However, when the image structure is broken by shuffling the pixels, we find that CNNs are no better at generalizing than MLPs. For this dataset, CNNs become substantially less compressible and hence our bounds show them generalizing worse than MLPs, see Figure 3 (center). MLPs do not suffer when this structure is broken since they never used it in the first place.

**Shuffled Labels.** When the structure of the dataset is entirely broken by shuffling the labels, the compressibility of the models (both for CNNs and MLPs) which fit the random labels is lost. Regardless of the subspace dimension used, our generalization bounds are all at $100\%$ error as shown in Figure 3 (right). It is not possible to fit the training data using low subspace dimensions, and when using a large enough dimension to fit the data, the compressed size of the model is larger than the training data and hence the generalization bounds are vacuous.

**Equivariance.** Designing models which are *equivariant* to certain symmetry transformations has been a guiding principle for the development of data-efficient neural networks in many domains [12, 13, 65, 66, 22, 35]. While intuitively it is clear that respecting dataset symmetries severely improves generalization, relatively little has been proven for neural networks [48, 21, 74, 4, 20]. We compress and evaluate rotationally equivariant ($C_8$) and non-equivariant Wide ResNets [66, 71] trained on MNIST and a rotated version of MNIST. As shown in Figure 5, the rotationally equivariant models are more compressible and provably generalize better than their non-equivariant counterparts when paired with a dataset that also has the rotational symmetry. See Appendix F for further details.

**Is Stochasticity Necessary for Generalization?** It is widely hypothesized that the implicit biases of SGD help to find solutions which generalize better. For example, Arora et al. [1] argue that there is no regularizer that replicates the benefits of gradient noise. Wu et al. [68], Smith et al. [61], and Li et al. [47] advocate that gradient noise is necessary to achieve state-of-the-art performance. In comparison, recent work by Geiping et al. [24] shows that full-batch gradient descent can match state-of-the-art performance, and Izmailov et al. [34] shows that full-batch Hamiltonian Monte Carlo sampling generalizes significantly better than mini-batch MCMC and stochastic optimization.

We train ResNet-18 and LeNet5 models on CIFAR-10 and MNIST, respectively, using full-batch and SGD with different intrinsic dimensionalities. We provide the training details in Appendix G. For MNIST with LeNet5, the best generalization bounds that we obtain are $11.55\%$ and $11.20\%$ using stochastic gradient descent (SGD) and full-batch training respectively. The best generalization bounds that we obtain for CIFAR-10 with ResNet-18 are $74.68\%$ and $75.3\%$ using SGD and full-batch training respectively. We also extend this analysis to SVHN to MNIST transfer learning with LeNet5 and obtain PAC-Bayes bounds of $9.0\%$ and $8.7\%$ using SGD and full-batch training respectively. These close theoretical guarantees on the generalization error for both SGD and full-batch training suggest that while the implicit biases of SGD may be helpful, they are not at all necessary for understanding why neural networks generalize well (see Appendix G).

**Double Descent.** Our bounds are also tight enough to predict the double descent phenomenon noted in Nakkiran et al. [54]. See Appendix I for a depiction of these experiments and a discussion of their significance.

## 7    Discussion

In this work, we constructed a new method for compressing deep learning models that is highly adaptive to the model and to the training dataset. Following Occam's prior, which considers shorter compressed length models to be more likely, we construct state-of-the-art generalization bounds across a variety of settings. Through our compression bounds, we show how generalization relates to the structure in the dataset and in the model, and we are able to explain aspects of neural network generalization for natural image datasets, shuffled pixels, shuffled labels, equivariant models, and stochastic training.

**Limitations.** Despite the power of our compression scheme and the ability of our bounds to faithfully describe the generalization properties of many modeling decisions and phenomena, we are scratching the surface of explaining generalization. Our compression bounds prefer models with a smaller number of parameters as shown in Appendix H, instead of larger models which actually tend to generalize better. While we achieved better model compression than previous works, it is unlikely that we are close to theoretical limits. Maybe through nonlinear parameter compression schemes we might find that larger deep learning models are more compressible than smaller models. Moreover, it is unclear how to relate the bounds of the compressed models to their uncompressed counterparts, perhaps leveraging ideas from Nagarajan and Kolter [52] and others investigating this question. Additionally, while our bounds show that the compressibility of our models implies generalization, we make no claims about the reverse direction. However, we believe that model compression and Occam's razor have yet untapped explanatory power in deep learning.

## Acknowledgements

We thank Pavel Izmailov and Nate Gruver for helpful discussions. This research is supported by NSF CAREER IIS-2145492, NSF I-DISRE 193471, NIH R01DA048764-01A1, NSF IIS-1910266, NSF 1922658 NRT-HDR, Meta Core Data Science, Google AI Research, BigHat Biosciences, Capital One, and an Amazon Research Award. This work is also supported in part through the NYU IT High Performance Computing resources, services, and staff expertise.

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
