# Supplementary Material for
# PAC-Bayes Compression Bounds So Tight
# That They Can Explain Generalization

## Appendix Outline

The appendix is organized as follows.

- In Appendix A, we report results for additional bounds for SVHN and ImageNet. We also report the compression size corresponding to our best bound values and compare it to the compression size obtained through standard pruning. Furthermore, in Appendix A.1 we prove why models cannot both be compressible and fit random labels.

- In Appendix B, we describe how optimization over hyperparameters like the intrinsic dimension impact the PAC-Bayes bound

- In Appendix C, we show how our PAC-Bayes bound benefit from transfer learning.

- In Appendix D, we discuss data-dependent priors and their effect on the subspace dimension optimization.

- In Appendix E, we detail our experimental setup including models, datasets, and hyperparameter settings for training and bound computation.

- In Appendix F, we provide a compression perspective to why equivariant models may be more desirable for generalization.

- In Appendix G, we further discuss how our through our PAC-Bayes compression bounds, we provide evidence that SGD is not necessary for generalization.

- In Appendix H, we ablate the model size and show how it impacts our bounds and compressibility, we identify the best performing size of models for our bounds.

- In Appendix I, we present our observations on double descent and their preditability from our PAC-Bayes bounds.

- In Appendix J, we expand our theoretical discussion and emphasize conceptual differences between our method and previous ones in the literature.

- Lastly, in Appendix K we provide licensing information on the datasets we use.

## A   Additional Results

In addition to the results reported in Table 2, we report the best bounds for SVHN and ImageNet-1k as well as the corresponding compressed size in Tables 3 and 4. In Table 3 we show how compressing the model via intrinsic dimension (ID) yields better results than standard pruning. In this table, we basically run our method but substitute ID with pruning and then proceed by quantizing the remaining weights and encoding them through arithmetic encoding. When pruning we used the standard iterative procedure following Han et al. [27], for the MNIST model we pruned 98.8% of the weights, for the FMNIST model 97.0% of the weights, for the SVHN model 98.8% of the weights and for both the CIFAR-10 and CIFAR-100 models we pruned 52.1% of the weights and stopped there as the accuracy dropped significantly if we kept pruning.

**Error bars on our bounds:** We re-run the bounds computation for 10 times and observe that the values are consistent. On average, we obtain $\pm 0.5\%$ variation in our bounds for models trained from scratch and $\pm 0.1\%$ variation for transfer learning models.

Table 3: Using our subspace method rather than pruning yields substantially higher compression ratios and hence tighter generalization bounds. We report our error bounds (%) and compressed size ($\mathbb{KL}$ (KB)), 1 KB = 8192 bits. First, we compress the model weights using ID, quantizing its values and then storing them through arithmetic encoding. We then report the bounds obtained by only switching ID to standard pruning. All results are data-independent and obtained with 95% confidence, i.e. $\delta = .05$.

| Dataset | ID + Quant + Arith | | Pruning + Quant + Arith | |
|---|---|---|---|---|
| | Err. Bound (%) | $\mathbb{KL}$ (KB) | Err. Bound (%) | $\mathbb{KL}$ (KB) |
| MNIST | **11.6** | 0.4 | 47.9 | 6.5 |
| + SVHN Transfer | **9.0** | 0.4 | | |
| FashionMNIST | **32.8** | 0.8 | 54.9 | 3.5 |
| + CIFAR-10 Transfer | **28.2** | 0.9 | | |
| SVHN | **36.1** | 1.3 | 74.4 | 4.3 |
| + ImageNet Transfer | **29.1** | 1.4 | | |
| CIFAR-10 | **58.2** | 1.2 | 100.0 | 57.8 |
| + ImageNet Transfer | **35.1** | 1.0 | | |
| CIFAR-100 | **94.6** | 4.1 | 99.9 | 50.7 |
| + ImageNet Transfer | **81.3** | 2.8 | | |

Table 4: Our PAC-Bayesian Subspace Compression Bounds with data-dependent priors compared to state-of-the-art PAC-Bayes non-vacuous data-dependent bounds. All results are obtained with 95% confidence, i.e. $\delta = .05$.

| Dataset | Err. Bound (%) | SoTA (%) |
|---|---|---|
| MNIST | **1.4** | 1.5 [59] |
| FashionMNIST | **10.1** | 38 [19] |
| SVHN | **8.7** | – |
| CIFAR-10 | **16.6** | 16.7 [59] |
| CIFAR-100 | **44.4** | – |
| ImageNet | **40.9** | – |

## A.1 Models that can fit random labels cannot be compressed

Our ability to construct nonvacuous generalization bounds rests on the ability to construct models which both fit the training data and are highly compressible. However, when the structure in the dataset has been completely destroyed by shuffling the labels, then we do not find that our models are compressible (shown in Figure 3 right). This is not just an empirical fact, but one that can be proven apriori: models which fit random labels cannot be compressed. While this result is a trivial consequence of complexity theory, we present an argument here for illustration.

**Almost all random datasets are incompressible**

When sampling labels uniformly at random, almost all datasets are not substantially compressible. Given a dataset $\mathcal{D} = \{x_i, y_i\}_{i=1}^n$ (where we are only considering the labels $y_i$, and conditioning on the inputs $x_i$), and denoting $|\mathcal{D}|$ as the length of the string of labels, the probability that a given dataset can be compressed to size $|\mathcal{D}| - c$ is less than $2^{-c+1}$. To see this, one must consider that there are only $\sum_{i=0}^{|\mathcal{D}|-c} 2^i \leq 2^{|\mathcal{D}|-c+1}$ programs of length $\leq |\mathcal{D}| - c$ (fewer still when restricting to self delimiting programs), and there are $2^{|\mathcal{D}|}$ possible datasets. Therefore averaging over all randomly labeled datasets the fraction which are compressible to less than or equal to $|\mathcal{D}| - c$ bits is at most $2^{|\mathcal{D}|-c+1}/2^{|\mathcal{D}|} = 2^{-c+1}$.

**A compressible model which fits the data is a compression of the dataset**

Let prior $P$ that includes a specification of the model architecture, and the model $h$ which outputs probabilities for each of the outcomes: $p(y = k \mid x_i) = h(x_i)_k$. We can decompose the (prefix) Kolmogorov complexity of the dataset (given the prior) as

$$K(\mathcal{D} \mid P) \leq K(\mathcal{D} \mid h, P) + K(h \mid P). \tag{5}$$

The term $K(\mathcal{D} \mid h, P)$ can be interpreted as a model fit term and upper bounded by the total negative log likelihood simply using the model probabilities as a distribution to encode the labels: $K(\mathcal{D} \mid h, P) \leq - \sum_i \log_2 h(x_i)_{y_i} + 1 = \text{NLL}(\mathcal{D} \mid h) + 1$.

Using the fact that almost all random datasets are incompressible, and choosing $c = 1 + \log_2(1/\delta)$, we have that with probability at least $1 - \delta$ over all randomly sampled datasets $K(\mathcal{D} \mid P) > |\mathcal{D}| - \log_2(1/\delta) - 1$. Plugging into Eq. (5) and rearranging, we have with probability $1 - \delta$,

$$K(h|P) \geq |\mathcal{D}| - \text{NLL}(\mathcal{D} \mid h) - \log_2(1/\delta) - 2, \tag{6}$$

In Figure 6 we plot the quantity $K(h \mid P) + \text{NLL}(\mathcal{D} \mid h)$ which represents the compressed size of the dataset achieved by our model (related to the minimum description length principle). We see that the value is considerably lower than the size of the dataset $|\mathcal{D}|$, emphasizing that real machine learning datasets such as CIFAR-10 have a very low Kolmogorov complexity and are very unlike those with random labels.

## B   Subspace Dimension Optimization and Hyperparameters in the Universal Prior

The smaller the chosen intrinsic dimension $d$, the more similar $\theta$ is to the initialization $\theta_0$ in Eq. (3). Consequently, that value of $\theta$ is more likely under the universal prior given the shorter description length. Note that in this prior, we condition on the random seed used to generate $\theta_0$ and $P$. As we optimize over different parameters such as the subspace dimension $d = 1, .., D$, and possibly other hyperparameters such as the learning rate, or number of quantization levels $L$, we must encode these into our prior and thus pay a penalty for optimizing over them. We can accomodate this very simply by considering the hypothesis $h$ as not just specifying the weights, but also specifying these hyperparameters: $h = (\theta, d, L, \text{lr})$, and therefore using the universal prior $P(h) = 2^{-K(h)}/Z$ we pay additional bits for each of these quantities: $K(h) \leq K(\theta \mid d, L) + K(d) + K(L) + K(\text{lr})$. If we optimize over a fixed number $H$ of distinct values known in advance for a given hyperparameter such as $L$, then we can code $L$ using this information in only $\log_2(H)$ bits. In general, we can also bound the dimensionalities searched over by the maximum $D$ so that $K(d) \leq \lceil \log_2 D \rceil$ in any case.

## C   Transfer Learning Bounds

We show the expanded results both with and without transfer learning in Table 3. When finetuning from ImageNet we use the larger EfficientNet-B0 models rather than the small convnet. Despite the fact that the model is significantly larger than the convnet or resnet models that we use to achieve the best bounds for from scratch training, the difference between the finetuned and pretrained models is highly compressible.

## D   Data Dependent Priors

We observe that when using data dependent priors, our optimization over the subspace dimension (and the complexity of the model used to fit the data when measured against the prior) favors very low dimensions and low KL values which we show empirically in Figure 4. Indeed, a large fraction of the data fitting is moved into fitting a good prior, particularly when the dataset fraction used to train the prior is large. When the prior is already fitted on the data, the final solution can have a very low complexity with respect to that prior without affecting data fit, and is encouraged to do so.

## E   Experimental Details

In this section we provide experimental details to reproduce our results.

### E.1   Model Training Details

We use a standard small convolutional architecture for our experiments, which we find produces better bounds than its ResNet counterparts. The architecture is detailed in Table 5, and we use $k = 16$ for experiments, but this value is ablated in Figure 6.

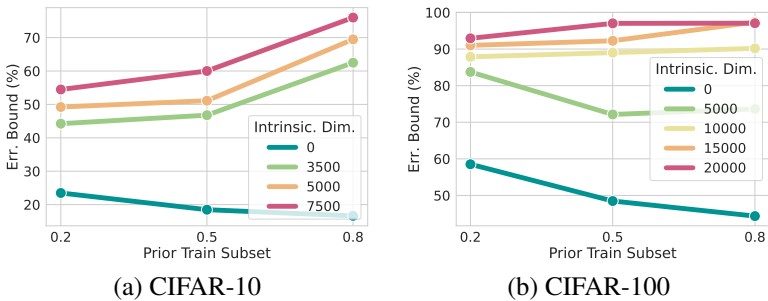

(a) CIFAR-10      (b) CIFAR-100

Figure 4: **Data-dependent bounds focus on fitting a good prior.** Our bounds using data dependent priors trained using varying fractions of the training dataset. We see that when using data dependent priors, lower intrinsic dimensionalities and lower KL models are favored by the bound.

**Stochastic training:** All models were trained for 500 epochs using the Adam optimizer with learning rate 0.001, except for ImageNet which was trained for 80 epochs with SGD using learning rate of 0.05 and weight decay of 0.00002. The model architectures for each dataset are listed below:

- MNIST [44] (+ SVHN [56] Transfer): LeNet-5 [44].
- FashionMNIST [69] (+ CIFAR-10 [38] Transfer): ResNet20 [28].
- SVHN: ConvNet (Table 5).
- SVHN + ImageNet Transfer: EfficientNet-B0 [63].
- CIFAR-10: ConvNet.
- CIFAR-10 + ImageNet Transfer: EfficientNet-B0 [63].
- CIFAR-100 [38]: ConvNet.
- CIFAR-100 + ImageNet [15] Transfer: EfficientNet-B0.

Table 5: Simple convolutional architecture we use to compute our bounds.

| ConvNet Architecture |
| --- |
| Conv(3,$k$), BN, ReLU |
| Conv($k$,$k$), BN, ReLU |
| Conv($k$,$2k$), BN, ReLU |
| MaxPool2d(2) |
| Conv($2k$,$2k$), BN, ReLU |
| Conv($2k$,$2k$), BN, ReLU |
| Conv($2k$,$2k$), BN, ReLU |
| MaxPool2d(2) |
| Conv($2k$,$2k$), BN, ReLU |
| Conv($2k$,$2k$), BN, ReLU |
| Conv($2k$,$2k$), BN, ReLU |
| GlobalAveragePool2d |
| Linear($2k$,$C$) |

**Full-batch training:** We train all models for 3000 epochs, use learning rates equal to 0.1 (MNIST + LeNet-5 and CIFAR-10 + ResNet-18) and 0.5 (CIFAR-10 + ConvNet), and a cosine learning rate scheduler that we warm-up for 10 epochs. We also clip the full gradient to have an $L_2$-norm of at most 0.25 before performing parameter updates in each epoch [24].

**Transfer Learning** All previous training details remain the same, except that $\theta_0$ from Eq. (3) is initialized from a pre-trained checkpoint instead of a random initialization. As typically done in literature, the final classification layer is replaced with a randomly initialized fully-connected layer to account for the number of classes in the downstream task.

## E.2 Bound Hyperparameter Optimization

As explained in Appendix B, we optimize the bound hyperparameters by considering that the hypothesis of interest $h$ specifies the hyperparameters in addition to the weights. Therefore, we pay bits back for the combination of hyperparemeters that we select. For example, if we are doing a grid search over 2 values of the quantization-aware training learning rate, 2 values of the intrinsic dimensionality values, 2 values of the quantization levels, and use k-means by default, then the number of bits that we pay is $\log_2(2 \times 2 \times 2) = 3$ bits.

**Optimizing PAC-Bayes bounds for data-independent priors:** Our PAC-Bayesian subspace compression bounds for data-independent priors have 4 hyperparameters that we list here-under alongside the possible values that we consider for each hyperparameter:

- The learning rate for the quantization-aware training, possible values: $\{0.001, 0.003, 0.005, 0.0001\}$.

- The intrinsic dimensionality, possible values: $\{0, 1000, 2500, 3000, 3500, 4000, 5000, 7500, 8000, 10000, 12000, 15000, 20000, 25000, 50000, 100000, 250000, 500000\}$, except for the ImageNet transfer learning which was conducted on the more limited range: $\{500, 1000, 2000, 3000, 4000, 6000, 8000\}$

- The number of quantization levels, possible values: $\{0, 7, 11, 30, 50\}$.

- The quantization initialization, possible values: $\{\text{uniform}, \text{k-means}\}$.

Note that we only use a subset of these hyperparameter values for some datasets, depending on the dataset size and other considerations. For all bound computations, we use arithmetic encoding and 30 epochs of quantization-aware training.

In Table 6, we summarize the hyperparameters corresponding to the data-independent bounds that we report in Table 2.

Table 6: Hyperparameters corresponding to our PAC-Bayesian Subspace Compression Bounds reported in Table 2 as well as SVHN and ImageNet to SVHN transfer learning with **data-independent priors**. All bound results are obtained with $95\%$ confidence, i.e. $\delta = .05$.

|  | Err. Bound (%) | Quant. Learning Rate | Intrinsic Dimensionality | Levels | Quant. Init. |
|---|---|---|---|---|---|
| MNIST | **11.6** | 0.005 | 1000 | 7 | Uniform |
| + SVHN Transfer | **9.0** | 0.005 | 1000 | 7 | Uniform |
| FashionMNIST | **32.8** | 0.005 | 2500 | 7 | Uniform |
| + CIFAR-10 Transfer | **28.2** | 0.005 | 2500 | 7 | Uniform |
| SVHN | **36.1** | 0.0001 | 3500 | 11 | Uniform |
| + ImageNet Transfer | **29.1** | 0.003 | 4000 | 7 | Uniform |
| CIFAR-10 | **58.2** | 0.0001 | 3500 | 7 | k-Means |
| + ImageNet Transfer | **35.1** | 0.003 | 3000 | 7 | Uniform |
| CIFAR-100 | **94.6** | 0.0001 | 10000 | 11 | k-Means |
| + ImageNet Transfer | **81.3** | 0.003 | 8000 | 7 | Uniform |

**Optimizing PAC-Bayes bounds for data-dependent priors:** In addition to the hyperparameters listed above, we also tune the hyperparameter corresponding to the subset of the training dataset that we use to train the prior on. We consider the following values for the subset of the training dataset: $\{20\%, 50\%, 80\%\}$.

In Table 7, we summarize the the hyperparameters corresponding to the data-dependent bounds that we report in Table 2. The best bounds are obtained for intrinsic dimensionality equal to 0, therefore no quantization is performed.

## E.3 Computational Infrastructure & Resources

Our computational hardware involved a mix of NVIDIA GeForce RTX 2080 Ti (12GB), NVIDIA TITAN RTX (24GB), NVIDIA V100 (32GB), and NVIDIA RTX8000 (48GB). The experiments were managed via W&B [5]. The total computational cost of all experiments (including the ones that do not appear in this work) amounts to $\approx 8000$ GPU hours.

Table 7: Hyperparameters corresponding to our PAC-Bayes bounds reported in Table 2 as well as SVHN and ImageNet with **data-dependent priors**. The best bounds are obtained for intrinsic dimensionality equal to 0, therefore no quantization is performed. All bound results are obtained with 95% confidence, i.e. $\delta = .05$.

|  | Err. Bound (%) | Training Subset (%) |
|---|---|---|
| MNIST | **1.4** | 50 |
| FashionMNIST | **10.1** | 80 |
| SVHN | **8.7** | 50 |
| CIFAR-10 | **16.6** | 80 |
| CIFAR-100 | **44.4** | 80 |
| ImageNet | **40.9** | 50 |

### E.4 Breaking Data and Model Structure Experiment

In this experiment we compared our generalization bounds derived for training convolutional networks and MLPs on standard CIFAR10, as well as when data structure is broken by shuffling the pixels or shuffling the labels. We trained for 100 standard epochs with batch size 128 and then another 50 epochs of quantization aware training in all cases. We use 7 quantization levels and uniform quantization initialization for all to simplify. When comparing against an MLP, we use a 3 hidden layer MLP with ReLU nonlinearities, and we feed in the images by flattening them into $3 \times 32 \times 32$ sized vectors. We use 150 hidden units in the intermediate layers of the MLP and choose $k = 46$ in the simple convolutional architecture described in Table 5 so as to match the parameter count (though slightly smaller models perform slightly better as ablated in Figure 6).

## F Equivariance

We conduct a simple experiment to evaluate the extent to which model equivariance has on the compressibility of deep learning models and the tightness of our generalization bounds. We use the rotationally equivariant $C_8$ WideResNet model from Weiler and Cesa [66] which has an 8-fold rotational symmetry, and we also use a non equivariant version of this model. The equivariant model has a depth of 10 and a widen factor of 4 yielding 1.451M parameters. We control for the number of parameters by adjusting the widen factor of the non equivariant model to 4.67 yielding 1.447M parameters.

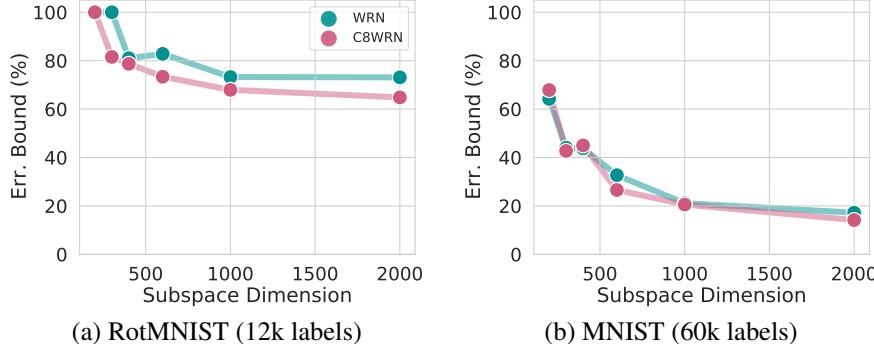

(a) RotMNIST (12k labels)  (b) MNIST (60k labels)

Figure 5: **Rotationally-equivariant models provably generalize better on rotationally-equivariant data.** Comparison of rotationally equivariant $C_8$ WideResNet vs ordinary WideResNet with the same number of parameters on (a) the rotationally equivariant RotMNIST dataset [42] and (b) the ordinary MNIST dataset. Both models are capable of fitting the data, but the equivariant model yields a more compressible solution when fitting the rotationally equivariant data than the non equivariant model, and hence yields a better generalization bound. (Note the difference in dataset size, RotMNIST has only 12K data points unlike MNIST)

We evaluate these models both on MNIST and the RotMNIST dataset [42] consisting of 12K training examples of rotated MNIST digits. As shown in Figure 5 (a), when paired with the rotationally

symmetric RotMNIST dataset, the rotationally equivariant model achieves better bounds and is more compressible than it's non equivariant counterpart despite having the same number of parameters. However, when this symmetry of the dataset is removed by considering standard MNIST, we see that the benefits of equivariance to the generalization bound and compressibility vs the WRN model dissapear.

# G    Full-Batch vs. Stochastic Training (SGD)

To further expand on the results that we present in Section 6, we study the impact of hyperparameters, namely the weight decay and the architecture choice, on the bounds obtained through full-batch (F-B) training. Table 8 summarizes these results and we provide the training and bound computation details in Appendix E. Our PAC-Bayes subspace compression bounds provide similar theoretical guarantees for both full-batch and stochastic training, suggesting that the implicit biases of SGD are not necessary to guarantee good generalization. Moreover, we see that the results are consistent for different configurations, which result in comparable bounds overall.

**Transfer learning using full-batch training:** We perform full batch training for transfer learning from SVHN to MNIST using LeNet-5 and the same experimental setup described in Appendix E. Our best PAC-Bayes subspace compression bounds for SVHN to MNIST transfer are $8.7\%$ and $9.0\%$ for full-batch and SGD training, respectively. This finding provides further evidence that good generalization of neural networks, and the success of transfer learning in particular, does not necessarily require stochasticity or additional flatness-inducing procedures to be achieved.

Finally, we note that we optimize over the same set of hyperparameters for the bound computation for both full-batch and stochastic training.

Table 8: Our PAC-Bayes subspace compression bounds obtained through full-batch (F-B) training for different configurations and datasets.

| Dataset | Architecture | Stochastic Err. Bound (%) | F-B Weight Decay | F-B Err. Bound (%) |
|---------|--------------|---------------------------|------------------|--------------------|
| MNIST | LeNet-5 | 11.6 | 0.01 | 12.5 |
| | | | 0.001 | **11.2** |
| | | | 0.005 | 12.0 |
| | | | 0.0001 | 11.7 |
| CIFAR-10 | ResNet-18 | 74.7 | 0.01 | 77.8 |
| | | | 0.001 | 76.3 |
| | | | 0.005 | 76.1 |
| | | | 0.0001 | **75.3** |
| CIFAR-10 | ConvNet | 58.2 | 0.01 | 65.8 |
| | | | 0.001 | 63.6 |
| | | | 0.0001 | **61.4** |

# H    Model Size vs. Compressibility

We perform an ablation to determine how the size of the model affects our generalization bounds. Using the fixed model architecture Table 5, we vary the width $k$ from 4 to 192. Using our subspace compression scheme, we find that the compression ratio of the model does increase with model size, however the total compressed size still increases slowly making our bounds less strong for larger models. For this paper, we find the sweet spot $k = 16$ is just above the point with equal number of parameters and data points.

We note that this finding leaves room for an improved compression scheme and generalization bounds which are able to explain why even larger models still generalize better. Curiously, when plotting to the total compressed dataset size ($K(h|P) + \text{NLL}$) using the model as a compression scheme, we find that the MDL principle which favors shorter description lengths of the data actually prefers larger models than our PAC-Bayes generalization bounds selects.

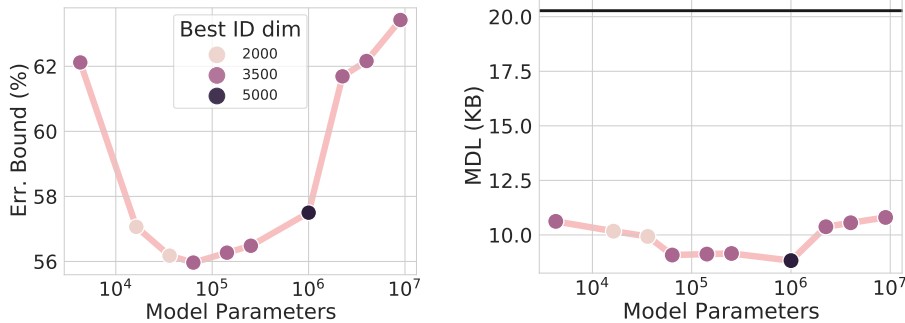

Figure 6: **Model size, compressibility, and MDL**. **Left:** Generalization error bound as a function of model size on the CIFAR10 dataset. The ID subspace dimension that achieves the best bound is shown by the color. In terms of our bound computation, the optimal number of parameters of the network is only slightly above the number of data points. **Right:** The total compressed size $(K(h|P) + \text{NLL})$ of the training dataset using our model as a compression scheme. While the raw labels have size 20.3KB (shown by the black line), the best model compresses the labels down to 8.6KB. Curiously, the compressed dataset size and hence the MDL principle favors larger models than our generalization bounds.

# I Double Descent

Under select conditions, we are able to reproduce the double descent phenomenon in our generalization bounds. In Figure 7 (right), we show that our bound exhibits a double descent similar to what we see in terms of the test error Figure 7 (left). The results we show in Figure 7 (right) are obtained for a fixed intrinsic dimensionality of 35000, but we observed that this middle descent consistently appears in our bounds plots for a given (fixed) intrinsic dimensionality where we select the best bound for each base width. However, we expect that extending the plot out to larger model widths the bound gets worse again as explained in Appendix H.

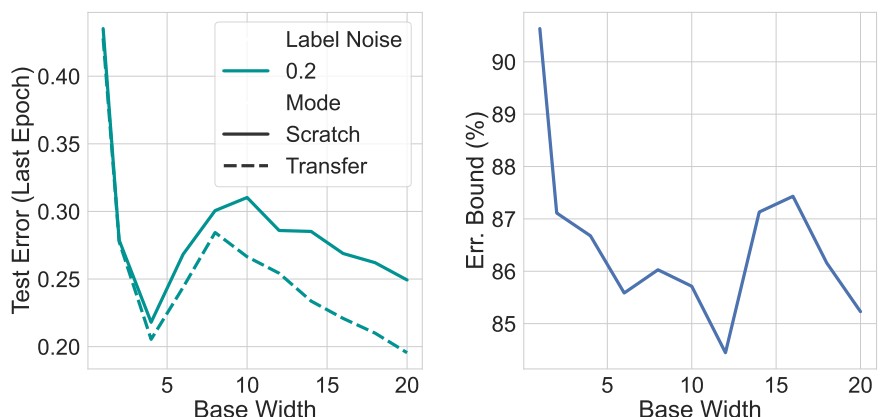

Figure 7: **Our bounds display a double descent as we increase the width. Left:** Double descent (in terms of the test error of the last epoch model) observed when varying the width of a ResNets-18 model to fit the CIFAR-10 dataset with label noise equal to 0.2. **Right:** Our bounds showing a similar *double descent* behaviour where the bound starts to worsen only to become better again at a later width. Here we can fix the intrinsic dimensionality to be equal to 35000 and we choose the best subspace compression bound for each base width.

## J  PAC-Bayes Bounds

### J.1  Catoni PAC-Bayes Bound

In our case, since neural networks achieve low training error, we focus on a bound like Catoni [7] which becomes tighter when $\mathbb{KL}\left(Q,P\right)$ is large. This is the same bound used in Zhou et al. [73].

**Theorem 1** (Catoni [7]). *Given a 0-1 loss $\ell$, a fixed $\alpha > 1$ and a confidence level $\delta \in (0,1)$ then*

$$\underset{\mathbb{E}}{\theta \sim Q}[R\left(f_\theta\right)] \leq \inf_{\lambda > 1} \Phi_{\lambda/N}^{-1}\left[\underset{\mathbb{E}}{\theta \sim Q}[\hat{R}\left(f_\theta\right)] + \frac{\alpha}{\lambda}\left[\mathbb{KL}\left(Q,P\right) + \log\frac{1}{\delta} + 2\log\left(\frac{\log\left(\alpha^2\lambda\right)}{\log\alpha}\right)\right]\right]$$

*holds with probability higher than $1 - \delta$ and where*

$$\Phi_\gamma^{-1}\left(x\right) = \frac{1 - e^{\gamma x}}{1 - e^\gamma}.$$

### J.2  Variable Length Encoding and Robustness Adjustment

In Zhou et al. [73], the authors assume a fixed length encoding for the weights. Given that the distribution over quantization levels is highly nonuniform, using a variable length encoding (such as Huffman encoding or arithmetic encoding) can represent the same information using fewer bits. While this choice gives significant benefits, it means that we cannot immediately make use of robustness adjustment from Zhou et al. [73], where the robustness adjustment comes from considering neighboring models that result from perturbing the weights slightly.

Revisiting the prior derivation in Zhou et al. [73], we show why the method used for bounding the KL does not transfer over to variable length encodings. In Zhou et al. [73], the prior used is

$$P = \tfrac{1}{Z} \sum_{S,Q,C} 2^{-(|S|+|C|+d\lceil \log L\rceil)}\mathcal{N}\left(\hat{w}\left(S,Q,C\right),\tau^2\right)$$

where $S$ denotes the encoding of the position of the pruned weights, $C$ denotes the codebook, $Q$ the codebook value that the weight take, $d$ the number of nonzero weights, $L$ the number of clusters, $\hat{w}$ the quantized weight and $\tau^2$ the prior variance. Note that $\hat{w}$ changes depending on $S,Q,C$ and also note that the fixed-length encoding can be seen in how we sum over $d\lceil \log L\rceil$ options. This prior is a mixture of Gaussians centered at the quantized values. With this choice of prior Zhou et al. [73] and setting the posterior to be also Gaussian centered at a quantized value, one can upper bound the KL with a computationally tractable term involving the sum over dimensions. Crucially, for their decomposition they use the fact that the size of the encoding $|Q|$ is $d\lceil \log L\rceil$, which is independent of the coding $Q$ and only dependent on the codebook $C$. Therefore they are able to upper bound the KL.

$$\mathbb{KL}\left(\mathcal{N}\left(\hat{w},\sigma^2 I_d\right), \sum_Q \mathcal{N}(\hat{w}(\hat{S},Q,\hat{C}),\tau^2)\right) = \sum_{i=1}^d \mathbb{KL}\left(\mathcal{N}\left(\hat{w}_i,\sigma^2\right), \sum_{j=1}^L \mathcal{N}\left(\hat{w}_j,\tau^2\right)\right)$$

due to the independence of the fixed length encoding to that of the values that each quantized value takes, see appendix in Zhou et al. [73]. This independence is broken for variable length encoding as the cluster centers and the values that each weight can take are interlinked. Thus, we cannot express and satisfactorily approximate the first high-dimensional KL term as a sum of one-dimensional elements that can be estimated through quadrature or Monte Carlo.

## K  Licences

MNIST [4] [44] is made available under the terms of the Creative Commons Attribution-Share Alike 3.0 license. FashionMNIST [5] [69] is available under the MIT license. CIFAR-10 and CIFAR-100 [38] are made available under the MIT license. ImageNet [6] [15] is the copyright of Stanford Vision Lab, Stanford University, Princeton University. SVHN [7] [56] is released for non-commericial use only.

---

[4] http://yann.lecun.com/exdb/mnist/
[5] https://github.com/zalandoresearch/fashion-mnist
[6] https://www.image-net.org/download
[7] http://ufldl.stanford.edu/housenumbers/