# OpenReview forum: "PAC-Bayes Compression Bounds So Tight That They Can Explain Generalization"
_NeurIPS.cc/2022/Conference — NeurIPS 2022 Accept_

### Official Review · Reviewer_GrpD · 2022-07-09

**Rating:** 4
**Confidence:** 2
**Soundness:** 2 fair
**Presentation:** 3 good
**Contribution:** 2 fair

**Summary:**

The paper proposes to apply model compression into an existing PAC-Bayes bound to obtain non-vacuous to obtain "better" prior from data (a portion of or the whole training dataset) to make the bound non-vacuous. The proposed method is based on the random subspace compression [38] that projects the high-dimensional parameter vector onto a lower-dimensional subspace with a fixed but random matrix projection, corresponding to the intrinsic dimension of the learning task. Such projected model parameter is then quantized to compress the model further.

**Questions:**

The contribution of the paper is not very clear to me. I wonder if it could be clarified further, especially differentiating the conventional random subspace projection and quantization with what are proposed in the current work.

Another suggestion is about the evaluation of other baselines in the literature. And since the current work includes two things: lower-dimensional projection and quantization, could the authors provide an ablation study to see the effect of using only the projection without the quantization. In such case, we could understand further how influence each of the proposed factors leads to the final result.

**Limitations:**

The authors explicitly stated the limitation where the current work only helps to unveil a part of the performance observed in deep learning models. There are certain aspects regarding to the generalization of deep learning models remained unexplained.

**Strengths And Weaknesses:**

**Strengths**

The main difference of the current work from the literature, to me, is the integration of model compression into existing PAC-Bayes bound to make the bound non-vacuous.

Another contribution of the current work is to present some ad-hoc methods based on either Kronecker product or the observation about Batch Normalization layers to improve the random subspace projection to find a better intrinsic dimension of the task of interest. It is then followed by a quantization technique to compress the model further to regularize the model of interest further according to the Occam's razor.

**Weaknesses**

The proposed random subspace projection in Section 4.1, despite improving the running time significantly, seems as an engineering technique to optimize [38].

Another suggestion is that the evaludation is quite simple with a limited number of baselines (both Tables 1 and 2). It seems that previous methods on non-vacuous PAC-Bayes bounds do not include results on several datasets, but it would make the paper more convincing if those methods could be reproduced and evaluated on the datasets of interest.

---

> ### Author Response · Authors · 2022-08-02
> **Response to Reviewer GrpD**
>
> Thank you for your review. Your primary concern about the paper seems to be that we have not adequately compared to other existing methods as baselines in our evaluation. We compared to the best achieved bounds that have been reported in the literature for each dataset, and the lack of entries reflects the fact that previously almost all bounds are vacuous for the more challenging image datasets. Following your suggestion though, we have rerun the methods of Zhou 2019 and Perez-Ortiz 2021 on FashionMNIST, CIFAR-10, and CIFAR-100 for better comparison. The best performing being Perez-Ortiz with error bounds of 46.5, 89.9, and 100 respectively, and we have updated the numbers in table 1 and 2. Notably, these methods yield near vacuous error bounds on the more difficult datasets CIFAR-10 and CIFAR-100. In contrast, our bounds 32.8, 58.2, and 94.6, improve considerably over state of the art, and that enables us through these bounds to investigate more subtle phenomena like the impact of equivariance, pixel shuffling, and convolution on generalization.
>
> In response to your question, we would like to clarify the contributions of the paper. We develop a framework for achieving the strongest generalization bounds for image classifiers to date. This framework leverages ideas from multiple different areas, connecting together the universal prior, intrinsic dimensionality, and learned quantization in both the parameters and the clusters. Traditional model compression techniques focus on minimizing inference compute and memory costs, and these methods do not achieve good results for generalization bounds. Critically, the use of training in random subspaces is novel in this context, and we are able to further improve on both the compute cost and performance of random subspaces by introducing a structured FiLM+Kron projector. Using these bounds we are able to investigate and understand various deep learning phenomena, drawing on the insight that structure in the data (as measured by Kolmogorov complexity) induces structure in the model, and the more structure the better the model generalizes.
>
> Regarding the impact of quantization, it is essential for generating non vacuous bounds. If instead of the 1-3 bits used per parameter with the current quantization scheme the weights are not quantized and the raw 32 bits per parameter for a float (or even 16 bits for a half) representations are used, all of the bounds we produce become vacuous. We also quantify the impact of replacing the random subspace training with more traditional pruning in appendix A, and pruning produces dramatically worse results, yielding only vacuous generalization bounds on CIFAR-10 and CIFAR-100 for example.
>
> Thank you again for your review and suggestions. We made a significant effort to address your questions, and would appreciate it if you would consider raising your score in light of our response. Please let us know if you have any additional questions we can address.

---

> > ### Author Response · Authors · 2022-08-06
> > **Re: Response to Reviewer GrpD**
> >
> > Thank you again for your thoughtful review. Does our response help address your questions? We would appreciate the opportunity to engage further if needed.

---

### Official Review · Reviewer_tugh · 2022-07-10

**Rating:** 4
**Confidence:** 3
**Soundness:** 2 fair
**Presentation:** 2 fair
**Contribution:** 2 fair

**Summary:**

The authors propose a new method for compressing deep learning models which is adaptive to the level of structure in the model and the training dataset. In their research, the authors follow the Occam razor paradigm with which they are able to explain why deep learning models generalize on structured data sets such as CIFAR-10.

**Questions:**

1. Can you type values for the missing values in the tables or explain why you can't do this?
2. Can an experiment be conducted to confirm the statistical effectiveness of the proposed approach?

**Ethics Review Area:**

["I don’t know"]

**Limitations:**

Yes

**Strengths And Weaknesses:**

Strengths:
- the proposed method is based on theoretical foundations,
- the subject of the research is interesting.

Weaknesses:
- poor comparison with other methods, many results of competing methods are not entered in the tables,
- there are no results confirming the statistical advantage of the proposed approach (it is worth running the method several times and presenting the mean and statistical deviations, it may perform a statistical test confirming the effectiveness of the method),
- the authors focused on the compression methods of the subspace dimension and the Occam razor paradigm, but there are many methods in the literature that reduce the number of model parameters in other ways. Couldn't you comment on them?  Here are some of them:
   * https://arxiv.org/pdf/1812.08928.pdf
   * https://openaccess.thecvf.com/content/CVPR2021/papers/Li_Dynamic_Slimmable_Network_CVPR_2021_paper.pdf
   * https://papers.nips.cc/paper/2017/file/a51fb975227d6640e4fe47854476d133-Paper.pdf
   * https://proceedings.neurips.cc/paper/2019/file/68b1fbe7f16e4ae3024973f12f3cb313-Paper.pdf
   * https://arxiv.org/pdf/1810.05331.pdf
- the presented charts are not sufficiently clearly described.

---

> ### Author Response · Authors · 2022-08-02
> **Response to Reviewer tugh**
>
> Thank you for your feedback. We address your questions below.
>
> We highlight that our paper makes different novel contributions. In addition to providing significantly stronger generalization bounds than prior work, enabled by a new efficient Kronecker projector and quantization schemes, we explore the effects of many design choices on the bounds, such as the role of equivariance, implicit regularization, and dataset structure. This analysis is qualitatively important: prior work has generally focused entirely on improving the bounds themselves, when ultimately the bounds will be most useful if they are prescriptive of model construction. Additionally, our methodology is not reliant on the use of data for achieving our bounds.
>
> *On missing values in the table*. As mentioned in our general comment to all reviewers, there are two main reasons for the crosses in Table 1. The first is that some methods like [19] or [55] can only be run on binary classification problems, thus they cannot provide a bound for multi-class datasets. The second is that the other methods such as [68], [20], [54] either provide vacuous bounds (as shown in the table we have now generated in the general comment) or like [20] cannot operate without data-dependent priors, whereas in Section 5.2 we explain why such bounds are incapable of explaining *why* neural networks generalize. Thank you for this point, and we have now updated our paper to include a clarification (see lines 34-35, Table 1). Additionally, we have filled-up Table 2 with the best performing alternative to us.
>
> *No results confirming statistical advantage*. On Appendix A (lines 593), we present the error bars for our bounds over 10 seeds.
>
> *Comment on compression method references*.  We want to emphasize here that the considerations for optimizing generalization bounds are fundamentally different from those in the compression literature at large, for example those which motivate many pruning methods.  In contrast to typical compression techniques, which are concerned with minimizing the number of FLOPs computed during a forward pass, reducing the memory footprint, or optimizing models for inference onboard specialized hardware, we are not concerned with efficient inference.  Instead, we are only concerned with achieving maximal compression of trained parameters, even if we introduce numerous untrained parameters (e.g. in the projection matrix) and ignoring the costs of compression and decompression.  Nonetheless, there are several papers which investigate properties related to our objective, one being the foundational paper (Han 2016) which explores learned pruning, nonuniform quantization levels, and Huffman coding, and (Choi 2017) further attempts to push these methods to the limit. In Appendix Section A, we compare our random subspace approach against more standard pruning methods from these last two references and find that our random subspace approach produces considerably tighter PAC-Bayes bounds. This is the case as our approach reduces the number of parameters much drastically than pruning.
>
> Thank you again for your review and suggestions. We made a significant effort to address your questions, and would appreciate it if you would consider raising your score in light of our response.  Please let us know if you have any additional questions we can address.

---

> > ### Author Response · Authors · 2022-08-06
> > **Re: Response to Reviewer tugh**
> >
> > Thank you again for your thoughtful review. Does our response help address your questions? We would appreciate the opportunity to engage further if needed.

---

### Official Review · Reviewer_9nYn · 2022-07-11

**Rating:** 7
**Confidence:** 3
**Soundness:** 4 excellent
**Presentation:** 4 excellent
**Contribution:** 4 excellent

**Summary:**

The paper demonstrates a novel combination of compression schemes that lead to significantly better generalization bounds derived from PAC-Bayes bounds. This approach is based on known bounds that were previously used by other works, but is combined with a more powerful compression scheme. The compression scheme is based on limiting the parameter space of model to a d-dimensional random subset represented by the addition of a random starting point in the parameter space plus the random project of a d-dimensional vector of learnable parameters. Such a representation was used in prior works to estimate the "inherent dimension" of a dataset / task. The d-dimensional vector is then optimized with quantization-aware training techniques, followed by quantization at the end of training. Finally, arithmetic coding is employed to retrieve the end compressed code of the classifier, which also includes additional hyperparameters on which the compression scheme is dependent (e.g., the seed). The estimated code-length is then plugged into the known bounds to result in the estimated generalization bounds.

The paper demonstrate how this method can obtain data-independent bounds that are sufficiently small to explain (to some extent) performance on MNIST, FMNIST, and CIFAR10, as well as additional phenomena, e.g., generalization of transfer learning, the importance of structure in the dataset for generalization of fully-connected vs. convolutional networks, and double descent. The paper also highlights some of the limitations of the current approach, for example, the estimated bounds do not correlate with the empirical observation that larger models generalize better.

**Questions:**

Think of the things where a response from the author can change your opinion, clarify a confusion or address a limitation. This can be very important for a productive rebuttal and discussion phase with the authors. ====

Are there any benefits to the Kroncker Projector beyond the runtime? Does it not harm model compression (in the "from scratch" scenario)? It would appear that Figure 1 (right) should also include a plot for the training accuracy when using the dense projector. While faster runtime is nice in practice, for theoretical exploration it seems secondary. If it leads to worse generalization bounds, then it will be detrimental to the main goal of the paper.

Also, some references are not properly cited, e.g., [2] above is cites by its arxiv number, not as an ICLR paper. I would advised making sure the citations are all updated and correct.

**Limitations:**

The authors gave a sufficient description of the limitations of their work.

**Strengths And Weaknesses:**

The paper gives strong evidence to support a long standanding conjecture that NN can be sufficiently compressed to a degree where compression-based bounds are close to their empirically observed accuracy. While not the first to follow this path or reach non-vacuous bounds, it obtains bounds that are far closer to real-world observations, and does so without relying on data-dependent bounds. The paper shows the limitations of data-dependent bounds that many prior works have focused on, that they are not as explanatory and are closer to bounds obtained by considering the estimation of the validation accuracy. The results on more complex tasks (CIFAR-10, CIFAR-100) are still far from what is desired, but the paper also shows the potential of this approach (its applicability for explaining a wide variety of phenomena) and indicates the area where researchers should focus. In addition, the paper is exceptionally well-written, clearly explaining PAC-Bayes bounds and their proposed compression scheme.

The main weakness of the paper is that many of its components are either wholly-based or lightly-derived from prior works, including the compression-based bounds and the individual components of the compression scheme. From that perspective, it is not as novel of an approach. It is also worth noting that the authors slightly misrepresent the degree to which using the "inherent dimension" factorization is novel in the context of model compression for estimating compression-based bounds. One could interpret Algorithm 1 in [1] as following such a factorization which appeared even prior to [2] that coined the term "inherent dimension". Of course the analyses and methods in that paper are far different from this submission. Regardless of the above, the specific combination of approaches (and their slight improvements) is novel, and it leads to the strong results mentioned above.

[1] - Arora et al., Stronger generalization bounds for deep nets via a compression approach, ICML 2018.

[2] - Li et al., Measuring The Intrinsic Dimension Of Objective Landscapes, ICLR 2018.

---

> ### Author Response · Authors · 2022-08-02
> **Response to Reviewer 9nYn**
>
> We appreciate your thorough analysis, feedback, and constructive review.
>
> Regarding your question on the improved parameter subspaces and their impact on the bounds, it is indeed the case that the primary motivation for the Kronecker projector is to reduce running time. However, it is not inferior to the Dense projector as it is simply another procedure for constructing a random subspace to explore. In fact, the Kronecker projector not only leads to no degradation of the generalization bound when training from scratch or with transfer learning but when equipped with the FiLM projector it achieves the best bounds even when compared to the Dense projector. This can be seen in the accuracy compression tradeoff in Figure 1 (left and right).
>
> We note your point on Algorithm 1 from [1] can in some sense be considered a precursor to our approach. There are some significant differences; [1] is applied layerwise (to each linear and convolution layer separately), after all training has been completed rather than in the training process, and with a more complicated form than just random projection for convolutional layers. We view [1] as a way of compressing neural networks similar to using low rank approximations of the weight matrices, which unlike intrinsic dimensionality is agnostic to how the compression impacts the training error. But still your point is relevant and we have added a sentence referencing in the related work section.
>
> *Fixing arxiv references*. Thanks for pointing this out. We have gone through the citations and made sure that they are up to date.
>
> As you highlighted: our work provides evidence that NNs can be sufficiently compressed, our method achieves tight bounds without the use of data-dependent priors, and our techniques have great potential for future bounds. Our bounds are sufficiently tight that they can help explain different properties of neural networks; such as the impact of equivariance, implicit regularization, and breaking structure in the data and model.

---

> > ### Author Response · Authors · 2022-08-06
> > **Re: Response to Reviewer 9nYn**
> >
> > Thank you again for your thoughtful review. Does our response help address your questions? We would appreciate the opportunity to engage further if needed.

---

### Official Review · Reviewer_1YMo · 2022-07-16

**Rating:** 6
**Confidence:** 3
**Soundness:** 3 good
**Presentation:** 2 fair
**Contribution:** 3 good

**Summary:**

Following a Pac-Bayes compression-based approach to generalization bounds developed in prior work, the paper leverages improved compression procedures to derive tight bounds for the compressed models, both in supervised and transfer learning settings. It also provides empirical evaluations of the bound upon variations of several factors known to impact generalization.

**Questions:**

Answering the above concerns (which justifies my low rating for now) would be a good starting point for a discussion.

Miscellaneous (minor points):

* Line 99: "relative entropy" and "entropy" could be named explicitly

* In the $KL$ bound below line 113: I believe $h$ should be $h^\ast$ in the right hand side expressions.

*  Fig 2 has no (a) or (b) as named on lines 290 and 296.

*  Role of implicit regularization: to dig deeper into the role of the optimizer, it would be interesting to see how the bound evolve during training compared to the actual test error.

**Limitations:**

The conclusion does include a discussion on the limitations of the results.

**Strengths And Weaknesses:**

The paper digs deeper into a nice approach that leverages a quantitative connection between compressibilty and generalization. Although the paper largely relies on the approach developed in prior work by Zhou et al (ref [59] in the paper), the practical improvements look substantial. I also find the discussion and empricial analysis in Section 6 particularly nice and promising.

My main reservation concerns a lack of clarity of the approach, which I feel can potentially lead to misinterpretations and misunderstandings. In fact I think these clarifications are important also for the reviewers so as to avoid misjudgements of the significance of the results.

From my understanding  from Section 3 and Algorithm 1 (please correct me if I'm wrong), the main idea (following Zhou et al) is to combine (i) a Pac-Bayes generalization bound applied to a pointwise posterior at the learned hypothesis and the the universal prior which penalizes predictors' minimum length; and (ii) improved compression schemes applied to the learned network weights.  This provides strong generalization guarantees for the compressed models. I believe the question of the relationship with the generalization gap for the uncompressed classifer is non trivial -- and left open here; I feel this is an important issue (which is addressed e.g in Nagarajan & Kolter (2019a)) which should be explicitly mentioned in the paper to avoid ambiguity. (I see that this limitation is explicitly mentioned and discussed in Arora et al 2018 for example, which also provides bounds for compressed models).

Note that this distinction is important conceptually: for example, recent analysis (e.g., Nagarajan & , Kolter (2019b),  Negrea et al (2019)) suggest that using suitable surrogates -- randomized or compressed -- versions of the learned predictors can circumvent the limitations of uniform convergence. From this point of view, it seems that the crux of the generalization problem is to explain generalization of the original predictor (e.g. by a suitable coupling that ties its  performance to that of the surrogate).

Regardless of clarify, a potentially more serious consequence of the above point is the soundness of the comparison with existing bounds. I believe it would not be methodologically fair to compare bounds on compressed models with bounds on uncompressed ones (just like it would not be fair to compare bounds on stochastic predictors  with bounds on deterministic ones). In fact, I would also be curious to see explicit comparisons with other standard existing (e.g., norm based) bounds applied to the compressed network. Could the authors comment on this point?

On the other hand, I do think that obtaining strong guarantees for well performing compressed models is very interesting. Now, if the goal is to obtain competitive compression models with guarantees,  I think the various tradeoffs that usually comes along with compression need to be more thoroughly investigated: in particular, how does the actual generalization performance  compare with the uncompressed network ? What are the computational costs (I would not agree that this issue is unimportant here as suggested on line 122-123) ? etc.

I think that the paper can potentially yield a  very nice contribution, provided the above ambiguity is correctly clarified and settled. For example, even without theoretical guarantees that tie the performances of the compressed and uncompressed model,  the bound for  the compressed model can still yield a powerful heuristic generalization measure for the uncompressed one. The results in Section 6 suggest that this is the case, but if this point of view is favoured, they would probably deserve a deeper and more thourough investigation, so as to compare well with prior work in the subject (in fact recent large scale correlation analysis revealed key challenges  in identifying actual causal factors for generalization (e.g. Jiang et al, 2019, Dziugaite et al 2020)).

References:

Nagarajan & Kolter (2019a). Deterministic PAC-Bayesian generalization bounds for deep networks via generalizing noise-resilience. ICLR 2019. https://arxiv.org/abs/1905.13344.

Nagarajan & Kolter (2019b). Uniform convergence may be unable to explain generalization in deep learning. NeuriPS 2019. https://arxiv.org/abs/1902.04742

Negrea, Dziugaite, Roy (2019). In Defense of Uniform Convergence: Generalization via derandomization with an application to interpolating predictors.  ICML 2010. https://arxiv.org/abs/1912.04265

Jiang et al (2019). Fantastic Generalization Measures and Where to Find Them. ICLR 2010. https://arxiv.org/abs/1912.02178

Dzuigaite et al (2020) In search of robust measures of generalization. NeurIPS 2020. https://arxiv.org/abs/2010.11924

---

> ### Author Response · Authors · 2022-08-02
> **Response to Reviewer 1YMo, part 1/2**
>
> Thank you for your constructive and detailed feedback. As you indicate, our bounds are for compressed models. We agree that being able to bound the performance of uncompressed models is of interest, and relating compressed and uncompressed models is a valuable direction for the community which is being explored. We have added references to these works in the limitations section. However, note that the papers exploring surrogate classes and linking compressed and uncompressed models that you provided are all asymptotic bounds, still providing vacuous bounds when applied to real neural networks. Meaningful bounds for standard uncompressed networks are as of yet still out of reach. The other methods which we compare to which yield non vacuous bounds such as Dziugaite and Roy 2017, Zhou et al 2019, Perez et al 2021 all apply to networks which are in some way compressed, rather than networks trained in the standard fashion.
>
> However, we emphasize that achieving strong and descriptive generalization bounds for a given dataset, regardless of whether the models are compressed or not, is a valuable and difficult objective itself. We view different generalization bound techniques as competing hypotheses for why deep learning works, and the extent to which a bound can in fact predict generalization serves as evidence for its respective hypothesis. Hence in our opinion, generalization bounds derived with different proof techniques on a fixed dataset can and should be compared directly, and we aim to and succeed in achieving the best generalization bounds overall on a given dataset, regardless of which category the bound method comes from.
>
> While models trained in the standard fashion (without quantization or projection to a low-dimensional subspace) may be highly compressible, existing tools from the broad literature on compression cannot adequately measure their compressibility, and this direction will be an important yet challenging long-term effort for the community.
>
> We can view our compressed training procedure as just simply another method to search for good settings of parameters in the original parameter space of the model which also achieve good generalization bounds. We emphasize that we are not trying to prove a bound over an entire hypothesis class (not even a surrogate class), but instead for the particular models found given the dataset, and we can use any method we choose for finding them. This is a somewhat different goal than Negrea et al 2019 and others you mention which prove a bound with uniform convergence, where members of a given hypothesis class all have a low generalization gap, and in this case the distinction between the surrogate class and the original hypothesis class is more important than in our setting. Many models within our hypothesis class provably do not generalize (e.g. Nagarajan & Kolter (2019b)), but we can prove that our models do.
>
> To your question, to the best of our knowledge, the most relevant norm based generalization measures for convolutional models are vacuous on the datasets we evaluate (MNIST, FashionMNIST, CIFAR10, CIFAR100) even on the compressed models, as seen in Hsu et al 2021 (Figure 1 and 2). Norm based methods generally assume a separation of the network into distinct linear layers and nonlinearities, and the bounds (only asymptotic anyways), apply to the norms of the individual layers which do not exist in a compressed form with our approach. So there is no obvious meaningful way that methods Spectral norm margins (Barlett et al 2017), Path norm (Neyshabur et al. 2017) or Fisher-Rao (Liang et al. 2019) can be applied to our compressed networks, and even if there were, these bounds are exponential in the depth of the network and vacuous on models of the size we consider.
>
> Hsu et al 2021. Generalization bounds via distillation. https://arxiv.org/pdf/2104.05641.pdf
> Bartlett et al 2017. Spectrally-Normalized Margin Bounds for Neural Networks. https://arxiv.org/pdf/1706.08498.pdf
> Liang et al 2019. Fisher-Rao Metric, Geometry, and Complexity of Neural Networks. https://arxiv.org/pdf/1711.01530.pdf
> Neyshabur et al 2017. Exploring Generalization in Deep Learning. https://arxiv.org/pdf/1706.08947.pdf

---

> > ### Author Response · Authors · 2022-08-02
> > **Response to Reviewer 1YMo, part 2/2**
> >
> >
> > *How does the actual generalization performance compare with the uncompressed network?* In order to maximize the strength of our generalization bounds (the objective in this paper), we employ very extreme compression ratios for the models leading to substantially reduced performance compared to models trained in a standard way. If a balance is desired between the empirical validation error and the constraint on the error from the generalization bound for e.g. certified models, one can strike a different balance on this compression - accuracy tradeoff. The extent of this tradeoff and typical accuracy of the networks can be seen from Figure 1 right. We notice in our experiments that it is very possible to achieve this trade-off. For instance for MNIST, we can achieve a 12.4% (best bound is 11.6%) data-independent error bound with a compressed and quantized model that has a test error 1.6% vs. 1.4% for the uncompressed model. Same for FashionMNIST, we can achieve 32.9% (best bound is 32.8%) data-independent error bound with a compressed and quantized model that has a test error 11.7% vs. 10.4% for the uncompressed model.
> > With regards to tradeoffs of compute time, explicitly training compressible models through our method has a computational cost of about ~5x that of training without the subspace or quantization. This cost is consequential in the context of optimizing the training of several models, yet it is not a limitation for elucidating different facets of generalization in deep learning (and we have reduced our training costs dramatically through our Kronecker subspace projector).
> >
> > Thank you for pointing out minor mistakes in the text which we have corrected. We also cited the works that you mentioned and think they are tightly related to our work. Overall, we think that your feedback has positively impacted our paper.  We hope we have been able to address your points, and that you will consider raising your score in light of our response. We would also be happy to further engage to answer other questions or concerns you may have.

---

> > > ### Author Response · Authors · 2022-08-06
> > > **Re: Response to Reviewer 1YMo**
> > >
> > > Thank you again for your thoughtful review. Does our response help address your questions? We would appreciate the opportunity to engage further if needed.

---

### Author Response · Authors · 2022-08-02
**General Comment to All Reviewers and Area Chair**

We thank all reviewers for their thoughtful and supportive feedback. Inspired by reviewer comments, we have run several experiments, which we include in the updated version of the paper. We begin with a general response and then address reviewers individually.

First, we would like to clarify an ambiguity in Table 1.  A cross in the table *does not* simply mean that the original paper corresponding to a method did not consider the dataset at hand, but rather a cross signifies that the method does not support multi-class problems, or that it is incompatible with the data-independent setting, or that it cannot yield competitive data-independent non-vacuous bounds. For example, the methods of Zhu et al. 2019 and Perez-Ortiz et al. 2021 do not achieve competitive data-independent non-vacuous bounds as shown in our new draft.  We have now enumerated the bounds generated by these methods – where feasible – in the table below and have updated our draft with a discussion to make this point clear.

We also would like to emphasize three key contributions of our work:

(1)  We propose a framework for achieving state-of-the-art PAC-Bayes generalization bounds. To this end, we derive novel compression techniques and quantization schemes which, in contrast to methods from the broader compression literature, are specifically geared towards tightening the PAC-Bayes bounds. Whereas mainstream compression literature focuses on efficient inference with limited memory bandwidth or onboard specialized hardware, our proposed framework instead minimizes the number of bits required to specify the parameters even if additional non-learned bits are required to specify a projection matrix and even if the method does not reduce inference speed because those properties do not factor into the bound.

(2)  Our bounds, unlike previous ones which strive to be nonvacuous are sufficiently tight that they can be used to analyze various properties of neural networks. For instance, our bounds allow us to measure the benefits of invariance in deep learning models and give insights into why deep neural networks can fit random labels yet generalize well on real labelings.

(3)  The significance of our focus on tight *data-independent* bounds is especially visible when contextualized within the recent PAC-Bayes generalization bound literature.  Other recent works have switched focus from data-independent bounds to data-dependent ones, which are easier to achieve but are significantly less informative about generalization as argued in Section 5.2 of our paper.  After all, bounds achieved using data-dependent priors do not indicate why exactly such a good prior was learned in the first place and thus seek to solve the mystery of generalization by introducing yet another mystery.  In contrast to the recent literature, our work shows that one can simultaneously build tight bounds while also using the more informative data-independent construction.  We hope that this achievement encourages the community to pursue the more explanatory data-independent direction in the future rather than their less interpretable data-dependent counterparts.

Our significantly tighter data-independent bounds and our application of these bounds to understand the generalization properties of deep neural networks are each significant contributions and are of broad interest not only to the generalization bound community but also to the sizable NeurIPS community interested in understanding why neural networks behave as they do.

| Reference | Binary MNIST | MNIST      | FMNIST       | CIFAR-10    | CIFAR-100  |
| ---       | ---          | ---        |  ---         | ---         | ---        |
| [18]      | 16.1         | x          |  x           | x           | x          |
| [58]      | 2.2          | x          |  x           | x           | x          |
| [71]      |              | 46         |  91.6        | 100         | 100        |
| [19]      |              | 11\*       |  38\*        | 23\*        | x          |
| [57]      |              | 21.7/1.5\* |  49.1        | 90.0/16.7\* | 100        |
| Ours      |              | 11.6/1.4\* |  32.8/10.1\* | 58.2/16.6\* | 94.6/44.4\*|

"*" indicates a data-dependent bound.

We clarify that [18] and [58] are fundamentally incompatible with multi-class classification. Moreover, [19] is completely reliant on data dependent priors and therefore cannot be used to obtain data-independent generalization bounds.

Please note that we moved the discussion on stochasticity for generalization and the double descent to appendix K. We intend to put it back to the main text in the camera ready version as we will be allowed to have a 10 page instead of a 9 page limit.

---

### Meta-Review · Area_Chair_WXjE · 2022-08-26

**Recommendation:** Accept
**Confidence:** Certain

**Metareview:**

After reading the submission and reviews, my understanding is that this submission combines the compression approach of Zhou et al with PAC-Bayes theory to obtain tight generalization bounds. While the approach is a mere combination of pre-existing approach, the insight that led to this method provides compelling results for generalization bounds and discussion with reviewers has strengthened the submission.
Therefore, I recommend this paper for acceptance.

**Award:**

No

---

### Decision · Program_Chairs · 2022-09-14

Accept